# Optimization design of top beam energy absorbing member of corrugated hydraulic support with three kinds of sandwich structures based on adaptive response surface method

**Zuen Shang, Qian Liu⊙\*, Jiyang Meng, Peng Yang, Xisheng Yu**

Liaoning Technical University, School of mechanical engineering, Fuxin, China

\* liuqianmxd@163.com

## Abstract

To enhance the energy absorption performance of support systems in response to the frequent rockburst issues arising from deep coal mining, a new energy-absorbing component with a corrugated core structure is proposed for use in hydraulic support beams. The study explores three core structures: triangular, trapezoidal, and sine. Three structures were designed and analyzed based on a mechanical model, and simulations were conducted using ANSYS software. The study investigates the variations in support reaction force and displacement, energy absorption and displacement, as well as buckling deformation, with a detailed analysis of the simulation results. Using the Adaptive Response Surface Method, structural parameter optimization was carried out with DESIGN EXPERT software for all three core structures. After obtaining the optimal configurations, experimental tests were conducted to verify the feasibility of these new structures as energy-absorbing components for hydraulic supports. The findings suggest that the optimal angle for the triangular corrugated unit is 30° with a thickness of 2.9mm. The trapezoidal corrugated unit exhibits optimal performance at a vertical angle of 34° and a thickness of 4.2mm. With regard to the quasi-sinusoidal corrugated unit, the optimal angle is 32° with a thickness of 3.2mm. Furthermore, in single-layer structures, the sinusoidal corrugated core structure exhibits the best performance in terms of energy absorption and displacement. The introduction of corrugated energy-absorbing components opens new avenues for structural innovation and performance optimization, offering valuable insights for designing anti-impact energy-absorbing structures.

## Introduction

As coal mining depths continue to increase, rockburst has become a significant challenge in deep mining operations [1–3]. To effectively mitigate rockburst hazards and

**Data availability statement:** All relevant data are within the paper and its Supporting Information files.

**Funding:** This study was funded by National Natural Science Foundation of China (Youth) Project: Dynamic 3D Scene Perception of Coal Mine Catastrophic Search and Rescue Robot Based on Multimodal Information Fusion; Project No.: 52404166 (2024, under research) Liaoning Provincial Department of Education Youth Improvement Project: 3D Environmental Perception of Coal Mine Disaster Scene Inspection Robot; Project No.: LJ212410147036 (Chaired in 2024, under research) Liaoning Provincial Department of Science and Technology Project: Research on Posture Control and Visual Cognition Algorithm of Coal Mine Rescue Robot Based on Origami Theory; Project No.: 2023-BS-204 (Principal Investigator in 2023)Liaoning Province Natural Fund Joint Fund; Project No.:2024-LH-0326. Funders provided strong support in study design, data collection and analysis.

**Competing interests:** The authors have declared that no competing interests exist.

reduce accident risks, robust and efficient support systems are essential. Some high-risk coal mines employ hydraulic supports with impact-resistant capabilities to provide advanced reinforcement for roadways [4]. However, without modifying the existing structural design of impact-resistant supports, optimizing the internal energy-absorbing components can significantly enhance energy dissipation efficiency. This not only mitigates the destructive effects of rockburst but also improves the adaptability of the support system, making it more suitable for complex engineering environments [5].

The current research on rockburst prevention support can be categorized into the following three areas:

(1) In the field of impact-resistant support theory, Y. S. Pan et al. [6] developed six design principles for impact-resistant support, based on statics and dynamics theory, and created a hydraulic support system with energy absorption functions for tunnels. Y. S. Pan et al. [7] established a "stress-surrounding rock-support" mechanical model for tunnels under rockbursts, proposed design principles for impact-resistant support considering both static and dynamic perspectives, and developed energy-absorbing support equipment, creating a three-tier energy-absorbing system that ensures stability and 3D energy absorption under rockburst conditions. M. S. Gao et al. [8], through strong-weak-strong structural models and numerical simulations, proposed an "internal support-external unloading" combination technique that effectively controlled the stability of deep rockburst tunnels and ensured the stability and safe recovery of surrounding rocks. Y. Z. Wu et al. [9] proposed the "decompression-support-protection" cooperative prevention and control principle, developing corresponding technologies and equipment that effectively reduce impact energy and improve tunnel surrounding rock stability.

(2) In proactive impact support, A. W. Wang et al. [10] summarized the design principles and experimental validation of a new type of impact-resistant energy-absorbing anchor rod (or cable), proving its excellent energy absorption, self-protection, and shock adaptability during tension and impact. Y. K. Fu et al. [11] developed high-impact toughness anchor rods, effectively enhancing the impact resistance of support structures. Zhao et al. [12] analyzed the anchoring effect of large deformation anchor rods and improved their compressive strength in rockburst events. Wang et al. [13] designed and studied a new type of Constant Resistance Energy Absorption (CREA) anchor, which effectively controls rockbursts, providing valuable insights for rockburst prevention in underground engineering.

(3) In the area of shock-resistant supports, Tang et al. [14] designed a hexagonal thin-walled energy-absorbing shock-resistant component that significantly enhances the impact resistance of the tunnel's advanced hydraulic supports. Tang et al. [15] introduced a hexagonal folded energy-absorbing shock-resistant component for mining, which effectively reduces peak load, enhances energy absorption, and improves stroke efficiency. When combined with existing hydraulic pillars, it significantly boosts impact resistance. Ma et al. [16] designed a chamfered tube shock-resistant support device and studied the influence of different panel angles on its energy-absorbing performance. The best energy absorption

was achieved with a panel angle of 156°.Ma et al. [17] designed a new type of energy-absorbing component for portal-type shock-resistant hydraulic supports, effectively protecting the hydraulic pillars from impact damage. [18] proposed an energy-absorbing hydraulic support design, which was shown to have superior energy absorption, stability, and support performance under impact loads compared to conventional supports. Wang Chunhua et al. [19] proposed a ribbed circular tube energy-absorbing shock-resistant component. Through simulations and experimental analysis, it was found that components with a Y-shaped arrangement and 8mm-thick ribs exhibited the best energy absorption and shock-resistance performance. Tian Liyong et al. [20] proposed a multi-cell thin-walled energy-absorbing component, which, after optimization, showed improved energy absorption and reduced support reaction fluctuations, enhancing the reliability of shock-resistant supports. An et al. [21] analyzed the mechanical performance of pre-folded tubes used for energy-absorbing support. By adjusting the stiffness of the concave sidewalls, the constant resistance performance can be improved, thereby increasing energy absorption. Jackowski et al. [22] examined the application of energy-absorbing structures in rear impact protection devices (RUPD) for motor vehicles, demonstrating their effectiveness in reducing impact energy and improving passenger safety during collisions. These structures are cost-effective and easy to implement. Shen Jiaxing et al. [23] designed an energy-absorbing structure using lattice materials with a negative Poisson's ratio, offering excellent energy absorption and support performance with stable load-bearing capacity. Wu Lili et al. [24] developed a composite sandwich panel made of GFPR top and bottom plates and a triangular cone lattice core, which provides high energy absorption efficiency and good economic viability. Zhang Jianzhuo et al. [25] proposed a solid-liquid coupling energy-absorbing component filled with emulsified fluid inside a sinusoidal corrugated tube, offering superior energy-absorbing components for collision protection equipment.

However, the variety of rockburst prevention structures or devices for coal mine tunnels remains limited, particularly in the area of lightweight and efficient rockburst support systems, which still offer significant potential for development and further research. To address this, this paper proposes a corrugated hydraulic support beam energy-absorbing component for tunnels, featuring three types of core structures. While honeycomb core structures offer significant improvements in specific energy absorption (SEA) and impact resistance, their mechanical performance can be further optimized through layer configuration, wall thickness adjustments, and velocity analysis [26]. However, corrugated cores provide a more cost-effective alternative, offering greater design flexibility and achieving high energy absorption while maintaining a lightweight structure. Therefore, this study investigates the application of corrugated structures in rockburst energy-absorbing advanced support hydraulic frames. The study involves model design, the construction of mechanical models, and optimization of the corrugated energy-absorbing components using response surface methodology. Additionally, simulation studies are conducted to examine their energy-buffering characteristics [27–30], followed by experimental validation.

## 1. Structural design scheme and energy absorption characteristic analysis method for energy-absorbing components

### 1.1. Design of corrugated energy absorbing component model

Based on the parameters and design requirements of the hydraulic support, the energy-absorbing component for the support beam with a corrugated structure and three core configurations is shown in Fig 1.

To ensure a secure connection to the hydraulic support top beam, the four corners of the top beam are welded with fixed pins to restrict the lateral and longitudinal movement of the component. The overall parameters for the single layer corrugated energy absorbing component designed in this study are shown in Table 1.

### 1.2. Analysis method of energy absorption components

Based on the influencing parameters of hydraulic supports and international evaluation standards for energy-absorbing components, this study primarily uses the following indicators to assess the effectiveness of the energy absorption.

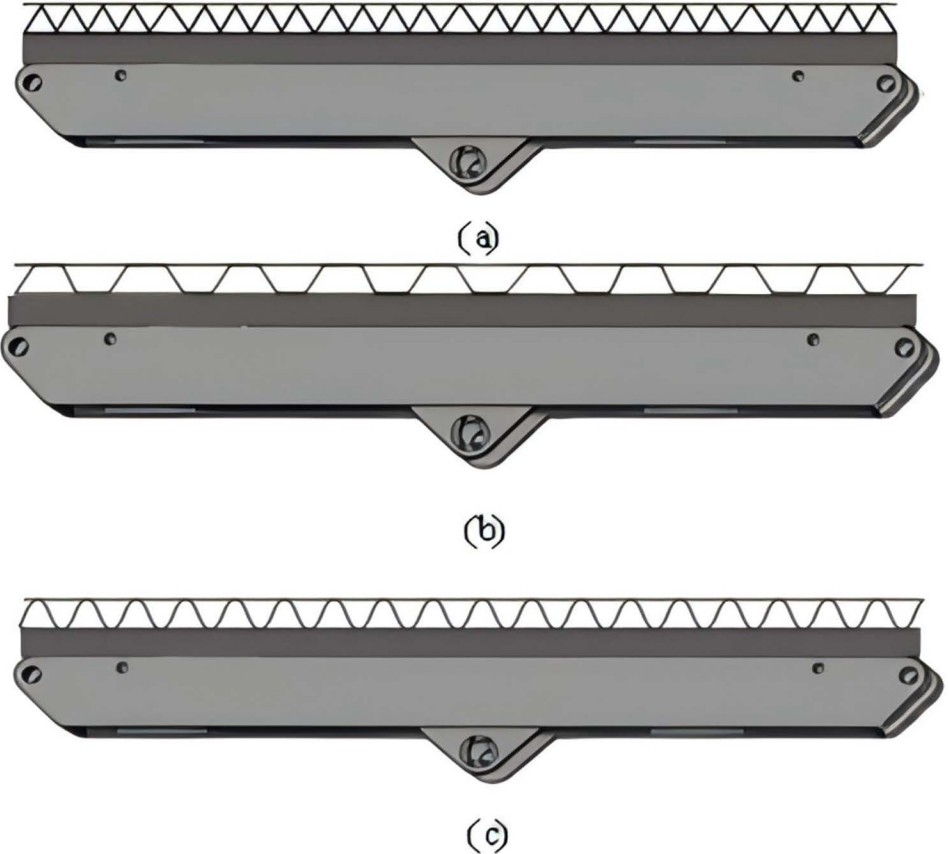

**Fig 1. Model of roof beam energy absorbing member of corrugated hydraulic support.** (a) Model of triangular energy absorbing member of roof beam of anti-scour hydraulic support. (b) Model of trapezoidal energy absorbing component of top beam of anti-impact hydraulic support. (c) Model of sinusoidal energy-absorbing component of the top beam of anti-impact hydraulic support.

**Table 1. Parameters of corrugated energy absorbing member with three kinds of sandwich structure.**

| Long | Wide | High | Braced Force |
| --- | --- | --- | --- |
| 5500mm | 920mm | 60mm | 5500kN |

Total Energy Absorbed $E_z$ /(J): The sum of the energy absorbed during the deformation of the component, which is one of the primary indicators of energy absorption efficiency.

$$E_z = M \int_0^{l_z} F_p \mathrm{d}l$$

(1.1)

Where $F_p$/(kN) is the platform load of the energy absorbing component; $l_z$/(mm) is the compression displacement; $M$ is a constant depending on the mechanical properties of the component material (usually between 0.8 and 1).

Specific energy absorption $E_{SEA}$/(kJ/kg): The ratio of the total energy absorbed during the effective compression displacement stage to the mass of the energy absorbing component. Specifically, it represents the amount of energy absorbed per unit mass of the structure during a collision. SEA is used to assess the energy absorption capacity of

materials or structures without causing excessive deformation. A higher SEA indicates better energy absorption capability, which is beneficial for impact resistance.[31] Let the mass of the energy-absorbing component be denoted as $M_z$/(kg).

$$E_{SEA} = E_z/M_z \tag{1.2}$$

Peak crushing load $F_{max}$/(kN): Indicates the maximum load capacity of the safety component.

$$F_{max} = n\sigma_s S_{max} \tag{1.3}$$

Where $n$ is the lateral deformation coefficient of the energy absorbing component ($n \leq 1$); $S_{max}$/(mm2) is the maximum effective area in the load direction; $\sigma_s$/(MPa) is the yield strength of the material.

Average Crush Load $F_m$/(kN)): Reflects the overall load level experienced by the component.

$$F_{\mathrm{m}} = \frac{E_z}{l_z} \tag{1.4}$$

Displacement Capability Coefficient $\gamma_s$: This measures the effectiveness of the energy absorbing component in the top beam of the hydraulic support during compression deformation. When the reaction force exceeds the initial peak crushing load and after hydraulic support safety valve failures, the deformation displacement is considered effective if the reaction force is less than 1.3 times the initial peak crushing load. The displacement coefficient thus reflects the displacement effectiveness of the component and indirectly indicates its plastic deformation capability.

$$\gamma_s = \frac{l_{SE}}{H} \tag{1.5}$$

Where $l_{SE}$/(mm) is the effective deformation displacement; $H$/(mm) is the vertical compression height of the energy absorbing component.

Load variation coefficient $\Delta$: During the compression deformation process of the energy absorbing component, a lower load variation coefficient indicates a more stable process. This stability, when applied to the top beam of the hydraulic support, indirectly reflects the component's ability to reduce shock and vibration to the hydraulic column. The lower the load variation coefficient, the better the damping performance.

$$\Delta = \frac{F_{max}}{F_m} \tag{1.6}$$

The compressive deformation process of the energy absorbing member is divided into several stages. During the transition from the elastic to the plastic deformation stage, or during unstable changes in the plastic deformation stage, sudden force changes may occur. The degree of load drop at this time reflects the impact on the load bearing device: the smaller the load drop, the less impact on the device, indicating better cushioning performance. Load Drop and Load Drop Coefficient are important indicators for evaluating the damping performance of energy absorbing components, as they indirectly reflect load variations.

Maximum Load Drop $\Delta F$/(kN):

$$\Delta F = F_{k\max} - F_{\min} \tag{1.7}$$

Load Drop Coefficient $\xi$:

$$\xi = \frac{\Delta F}{F_{\min}} \tag{1.8}$$

Where $F_{k\,max}$/(kN)is the maximum load within the effective deformation displacement; $F_{min}$/(kN) is the minimum load after the drop.

The peak crushing load should be maximized within a reasonable load-carrying range; for components of the same size, greater energy absorption after compression is preferable; a smaller load variation coefficient indicates better damping performance. For wide application in practical engineering, energy absorbing components should meet the requirements of simple structural design and low material cost.

## 2. Construction of mechanical model for the sandwich of corrugated energy absorbing member

By developing a mechanical model, the relationship between the thickness of the three corrugated cores and their parameters is analyzed, resulting in corresponding theoretical formulas that provide theoretical support for simulations and experiments.

### 2.1. Design and analysis of mechanical model of triangular corrugated core

A mathematical model is set up for the triangular corrugated sandwich structure as shown in Fig 2. The corrugated structure is subjected to a total pressure of $F_z$/(kN), with component dimensions of $l$/(mm) in length and $h$/(mm) in height. The vertical angle of the triangular corrugated core is $\beta$/ (°), and the force on a single corrugated chip is $F_s$/(kN). This structure can be simplified to a case where a concentrated force acts on the top edge. Due to the symmetry of the component, half of the triangular corrugated core is used as the object of mechanical analysis.

Based on Fig 2, the bending moment is assumed to be an additional unknown force. The thickness of the corrugated sandwich is calculated to meet the required conditions.

The unit bending moment $\overline{M}_1$/(N·m) and the load bending moment $M_P$/(N·m) of the triangular corrugated sandwich structure are respectively:

$$\overline{M}_1 = 1 \tag{2.1}$$

$$M_P = -\frac{F_s}{2}x\sin\beta \left( 0 \leq x \leq \frac{h}{\cos\beta} \right) \tag{2.2}$$

The effective deformation displacement coefficient $\delta_1$ and the free term $\Delta_1$ are:

$$\delta_1 = \int \frac{\overline{M}_1^2 ds}{EI} = \frac{h}{EI\cos\beta} \tag{2.3}$$

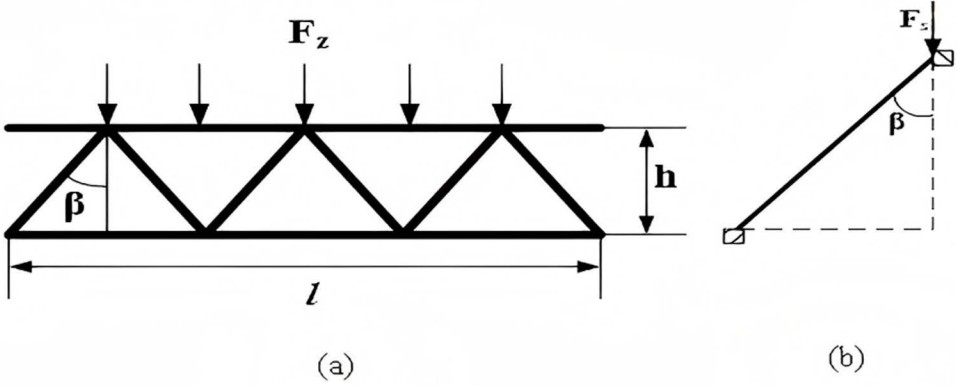

(a)　　　　　　　　(b)

**Fig 2. Numerical model of triangular corrugated core.** (a) Force on triangular corrugated core structure. (b) Force on a single triangular corrugated chip.

Where $E$/(GPa) is the modulus of elasticity of the material and $I$/(N·m) is the moment of inertia.

$$\Delta_1 = \int \frac{\overline{M_1}M_p}{EI}ds = -\frac{F_s h^2}{2EI}\sin\beta \tag{2.4}$$

The two forms of expression for the ultimate bending moment $M_s$/(N·m) are:

$$M_s = \overline{M_1}\frac{\Delta_1}{\delta_1} \tag{2.5}$$

$$M_s = \frac{1}{2}\sigma_s d m_s{}^2 \tag{2.6}$$

$$F_s = \frac{2F_z h \tan\beta}{l} = k\frac{\sigma_s d m_s{}^2}{2h\tan\beta} \tag{2.7}$$

Derived from equations 1.8 to 2.6:

$$m_s{}^2 = \frac{4F_z(h\tan\beta)^2}{lk\sigma_s d} \tag{2.8}$$

Where $d$/(mm) is the width of the energy absorbing component, $m_s$/(mm) is the thickness of the triangular corrugated core support plate, $s$/(mm2) is the cross-sectional area of the triangular corrugated core support plate, and $k$ is the material's yield strength coefficient of the material.

## 2.2. Design and analysis of mechanical model of trapezoidal corrugated core

Mathematical model structure of a trapezoidal corrugated core:

A mathematical model for trapezoidal corrugated core components is established. The study focuses on one half of a single trapezoidal corrugated core and creates a corresponding mechanical mathematical model.

The effective deformation displacement coefficient $\delta_1$ and the free term $\Delta_1$ of a single trapezoidal corrugated core are derived from Fig 3 and the related theoretical models as follows:

$$\delta_1 = \int \frac{\overline{M_1^2}ds}{EI} = \frac{1}{EI}\left(\frac{d_1}{2} + \frac{h}{\cos\alpha}\right) \tag{2.9}$$

$$\Delta_1 = \int \frac{\overline{M_1}M_p}{EI}ds = -\frac{1}{EI}\left[F_s\sin\alpha\left(\frac{h}{\cos\alpha}\right)^2 + \frac{F_s d_1{}^2}{48}\right] \tag{2.10}$$

$$F_s = \frac{2F_z(2h\tan\alpha + d_1)}{l} \tag{2.11}$$

The thickness of the trapezoidal corrugated core plate is determined using the relevant equations, namely 2.9, 2.10 and 2.11, in conjunction with the two expressions for the ultimate bending moment.

$$m_t{}^2 = \frac{2F_z(2h\tan\alpha + d_1)\left[2\sin\alpha(\frac{h}{\cos\alpha})^2 + \frac{d_1{}^2}{24}\right]}{\sigma_s dL(\frac{d_1}{2} + \frac{h}{\cos\alpha})} \tag{2.12}$$

The location of the aforementioned item is as follows: The top width of the trapezoidal corrugated core structure is represented by $d_1$/(mm), while the bottom width is represented by $d_2$/(mm). The angle between the trapezoidal corrugated core

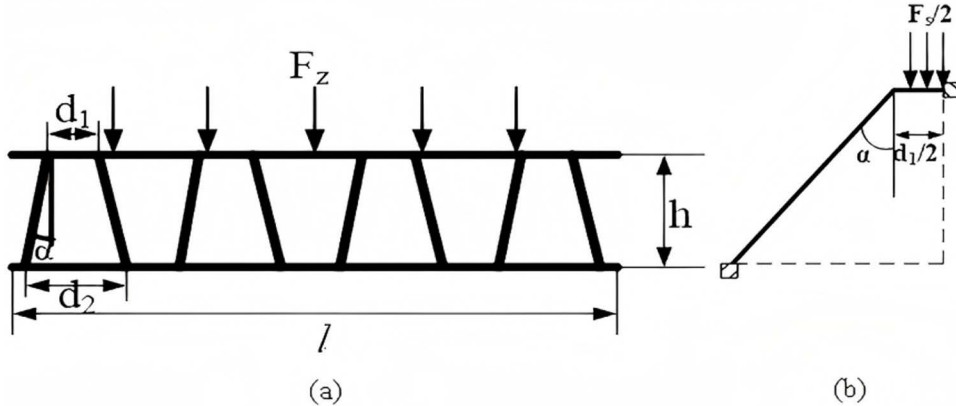

**Fig 3. Numerical model of trapezoidal corrugated core.** (a) Stress model of trapezoidal corrugated core structure. (b) Stress model of single trapezoidal corrugated chip.

and the vertical direction is represented by $\alpha/$ (°), and the thickness of the support plate for the trapezoidal corrugated core is represented by $m_t/$(mm).

### 2.3. Design and analysis of mechanical model of sinusoidal corrugated core

A preliminary three-dimensional model of the sinusoidal corrugated structure has been developed, and a mathematical model has been established based on this model, as illustrated in Fig 4. This model represents the overall force $F_z$. Furthermore, the force $F_1$ applied by each corrugated core plate to the top plate is also illustrated. As a consequence of the symmetry inherent to the mathematical model, the force model for a single corrugated core plate is illustrated in Fig 4.

The equation for the sinusoidal-like curve function is as follows:

$$f(x) = a\sin(bx) \tag{2.13}$$

Where: $a$ is a constant related to the height of the corrugated structure; $b$ is the varying constant.

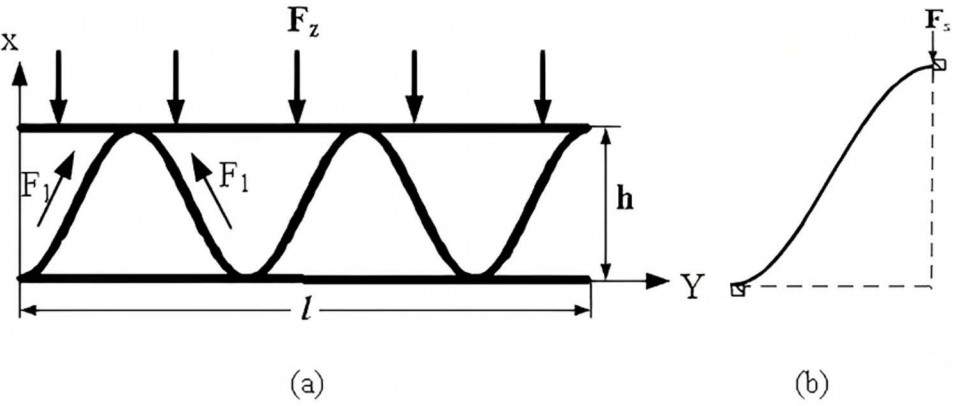

**Fig 4. Numerical model of sinusoidal corrugated sandwich.** (a) Stress model of sinusoidal corrugated core structures. (b) Stress model of single sinusoidal corrugated chip.

The angle $\eta/(°)$ between the small element and the horizontal direction is:

$$\eta = \arctan(ab \cos bx) \tag{2.14}$$

Upon performing the derivative calculation on the sinusoidal function curve, the following result is obtained: $0 \leq \eta \leq 45°$

The bending moment at the small element under the applied load can be expressed as follows:

$$M_P = -\frac{F_s}{2}x\sin\eta \left(0 < x \leq \frac{\pi}{b}\right) \tag{2.15}$$

The effective deformation displacement coefficient $\delta_1$ and the constant term $\Delta_1$ are provided as follows:

$$\delta_1 = \int \frac{\overline{M_1^2}ds}{EI} = \frac{1}{EI}\int_0^{\frac{\pi}{b}} \sqrt{a^2b^2\cos^2 bx + 1}\,dx \tag{2.16}$$

$$\Delta_1 = \int \frac{\overline{M_1}M_p}{EI}ds = -\frac{1}{EI}\int_0^{\frac{\pi}{b}} \frac{F_s\pi}{2b}dx \tag{2.17}$$

The two expressions for the ultimate bending moment are as follows:

$$M_s = \overline{M_1}\frac{\Delta_1}{\delta_1} \tag{2.18}$$

$$M_s = \frac{1}{2}\sigma_s dm_z{}^2 \tag{2.19}$$

The force acting on a single corrugated core plate is a function of the plate's geometry and the loading conditions:

$$F_s = \frac{2\pi F_z}{bl} \tag{2.20}$$

The thickness of the sinusoidal-like corrugated core plate $m_z/(mm)$ is determined by simultaneously solving Equations 2.16, 2.17, 2.18, 2.19 and 2.20.

$$m_z{}^2 = \frac{2F_z\pi^3}{\sigma_s db^3 l \int_0^{\frac{\pi}{2b}} \sqrt{a^2b^2\cos^2 bx + 1}\,dx} \tag{2.21}$$

## 3. Optimization and analysis of structural parameters of corrugated energy-absorbing member based on response surface method

For parameter optimization, it is essential to identify design parameters for different corrugated unit models and to establish corresponding parametric equations between different variables. To facilitate the creation of parametric equations, this study identifies parameters for different corrugated core structures by analyzing the mechanical properties of triangular, trapezoidal and sinusoidal structures. The parameters selected as variables included (top width: 0 mm for triangular and sinusoidal, 1 mm for trapezoidal), the angle β between the inclined side and the vertical direction (as shown in Fig 5, where the angle between the sinusoidal peak-valley junction and the vertical is also defined as β), and the thickness t of the corrugated core.

Based on single-factor finite element modelling, this study analyses the combined effects of different influencing factors on the energy absorbing component. The three main factors influencing the energy absorption metrics are d、β and t, which interact with each other. The study investigates how these multi-factor structural parameters influence the performance of the energy absorbing component. An orthogonal experimental design is used, followed by corresponding finite element analyses. Using energy absorption and specific energy absorption as the evaluation criteria, the data are presented

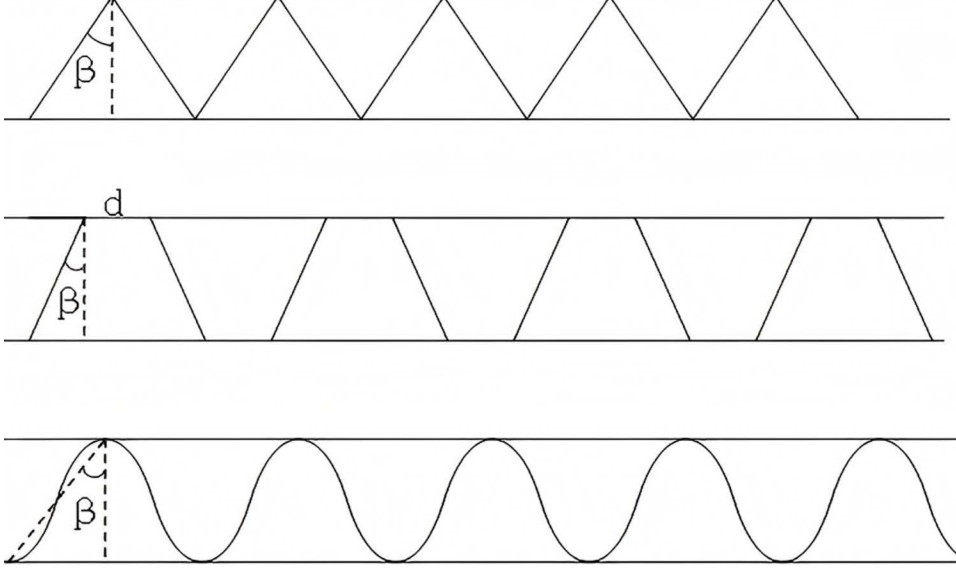

**Fig 5. Parameter specification diagram.**

**Table 2. Response surface factor level table.**

| Considerations | Coding level | | |
|---|---|---|---|
| | -1 | 0 | 1 |
| d(mm) | / | 0 | 0 |
| β(°) | 20 | 35 | 50 |
| t(mm) | 1.6 | 4.5 | 7.4 |

in Table 3. Preparatory work for response surface analysis is carried out, resulting in the response surface factor values shown in Table 2.

Based on the data in Table 3, the response surface was generated using Design Expert software to fit the curve. Fig 6 presents a response surface illustrating the interaction between d, β, t, and energy absorption. Fig 7 depicts a response surface representing the interaction between d, β, t, and specific energy absorption.

In optimization problems, it is essential to define the variables, clarify the objective function and set the constraints. By constructing an optimal design model and selecting appropriate variables $x$, the objective function can be optimized. The mathematical model for this process can generally be expressed in the following form:

$$\begin{cases} max(or\ min)f(x) \\ s.t\ x^l \leq x \leq x^u \end{cases}$$ (3.1)

$f(x)$ represents the objective function, $x$ represents the design variables of the optimization model, $x^l$ denotes the lower bounds of the n design variables, and $x^u$ denotes the upper bounds of the n design variables.

To ensure that the energy absorbing components of the hydraulic support maximized both effective displacement and energy absorption, energy absorption and specific energy absorption were chosen as objective functions. Polynomial curve fitting was performed using Design Expert software, resulting in the following fitted curves:

$$S_E(d, \beta, t) = 2.26 - 0.16d + 0.71\beta + 1.08t + d\beta - 1.19dt - 0.33\beta t$$ (3.2)

**Table 3. Experimental design and results of response surface.**

| d(mm) | β(˚) | t(mm) | Specific absorption energy (kJ/kg) | Energy absorption (kJ) |
|---|---|---|---|---|
| 0 | 20 | 1.8 | 1.01 | 0.4 |
| 0 | 22 | 2.0 | 1.18 | 0.53 |
| 0 | 24 | 2.2 | 1.38 | 0.69 |
| 0 | 26 | 2.4 | 1.50 | 0.83 |
| 0 | 28 | 2.6 | 1.67 | 1.02 |
| 0 | 30 | 2.9 | 1.96 | 1.35 |
| 0 | 32 | 3.1 | 1.91 | 1.45 |
| 0 | 34 | 3.3 | 1.94 | 1.61 |
| 0 | 36 | 3.6 | 1.80 | 1.68 |
| 0 | 38 | 3.9 | 1.83 | 1.87 |
| 0 | 40 | 4.1 | 1.84 | 2.04 |
| 0 | 42 | 4.5 | 1.81 | 2.29 |
| 0 | 44 | 4.8 | 1.72 | 2.41 |
| 0 | 46 | 5.1 | 1.67 | 2.58 |
| 0 | 48 | 5.5 | 1.64 | 2.84 |
| 0 | 50 | 5.9 | 1.54 | 2.98 |
| 1 | 20 | 2.2 | 0.77 | 0.37 |
| 1 | 22 | 2.5 | 0.71 | 0.5 |
| 1 | 24 | 2.8 | 0.82 | 0.68 |
| 1 | 26 | 3.0 | 0.96 | 0.94 |
| 1 | 28 | 3.3 | 1.18 | 1.28 |
| 1 | 30 | 3.6 | 1.28 | 1.54 |
| 1 | 32 | 3.9 | 1.25 | 1.65 |
| 1 | 34 | 4.2 | 1.44 | 2.08 |
| 1 | 36 | 4.5 | 1.39 | 2.25 |
| 1 | 38 | 4.8 | 1.29 | 2.48 |
| 1 | 40 | 5.2 | 1.39 | 2.67 |
| 1 | 42 | 5.6 | 1.31 | 2.77 |
| 1 | 44 | 6.0 | 1.26 | 2.94 |
| 1 | 46 | 6.4 | 1.24 | 3.14 |
| 1 | 48 | 6.9 | 1.18 | 3.25 |
| 1 | 50 | 7.4 | 1.15 | 3.38 |
| 0 | 20 | 1.6 | 0.76 | 0.14 |
| 0 | 22 | 1.9 | 0.96 | 0.24 |
| 0 | 24 | 2.1 | 1.13 | 0.35 |
| 0 | 26 | 2.4 | 1.9 | 0.8 |
| 0 | 28 | 2.7 | 1.86 | 0.93 |
| 0 | 30 | 2.9 | 1.65 | 0.98 |
| 0 | 32 | 3.2 | 1.91 | 1.4 |
| 0 | 34 | 3.5 | 1.84 | 1.61 |
| 0 | 36 | 3.7 | 1.82 | 1.79 |
| 0 | 38 | 4 | 1.74 | 1.94 |
| 0 | 40 | 4.3 | 1.71 | 2.36 |
| 0 | 42 | 4.6 | 1.68 | 2.42 |
| 0 | 44 | 4.9 | 1.64 | 2.54 |
| 0 | 46 | 5.2 | 1.58 | 2.69 |
| 0 | 48 | 5.5 | 1.53 | 2.97 |
| 0 | 50 | 5.8 | 1.49 | 3.01 |

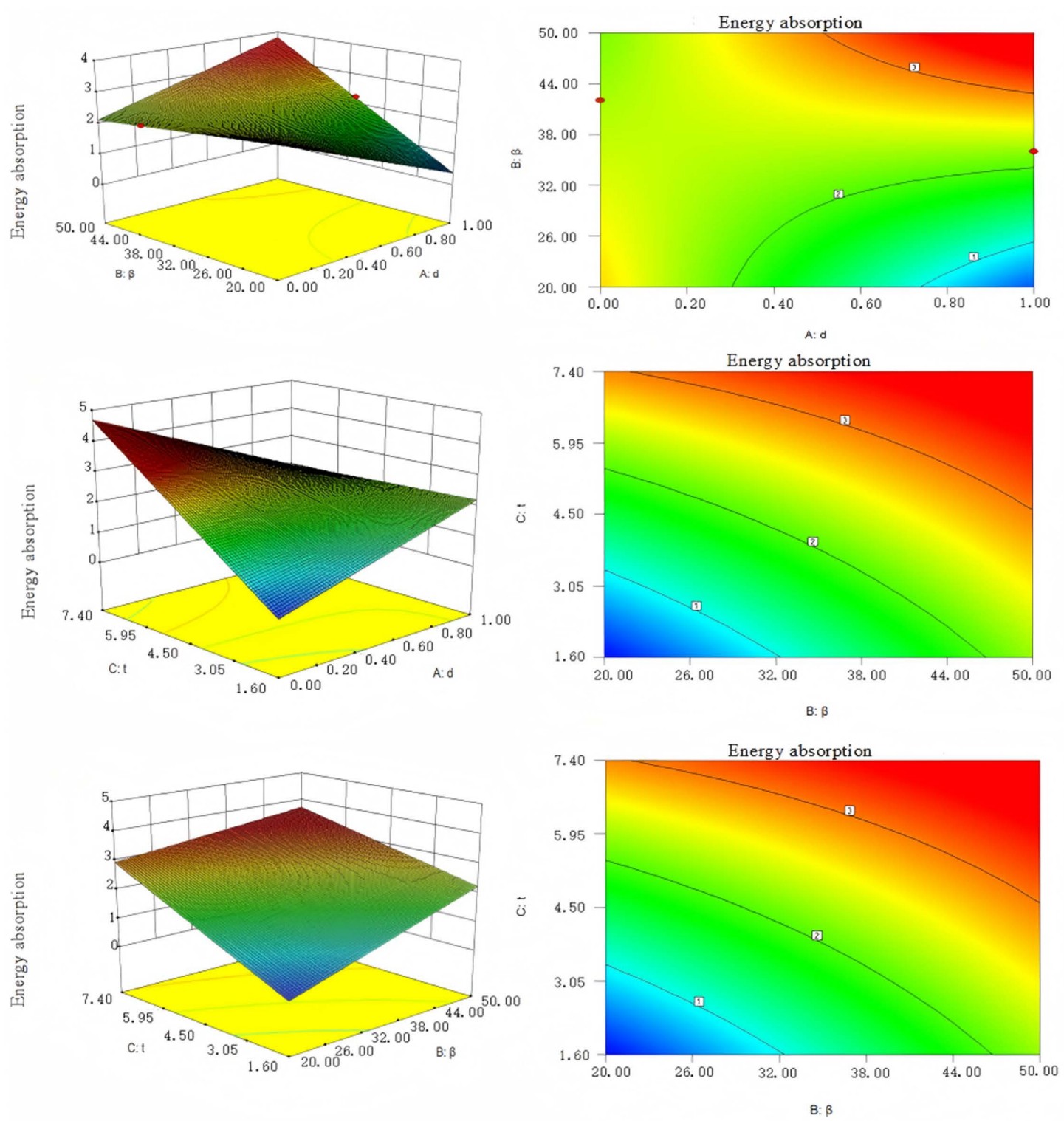

**Fig 6. The response surface of d, β and t on energy absorption.**

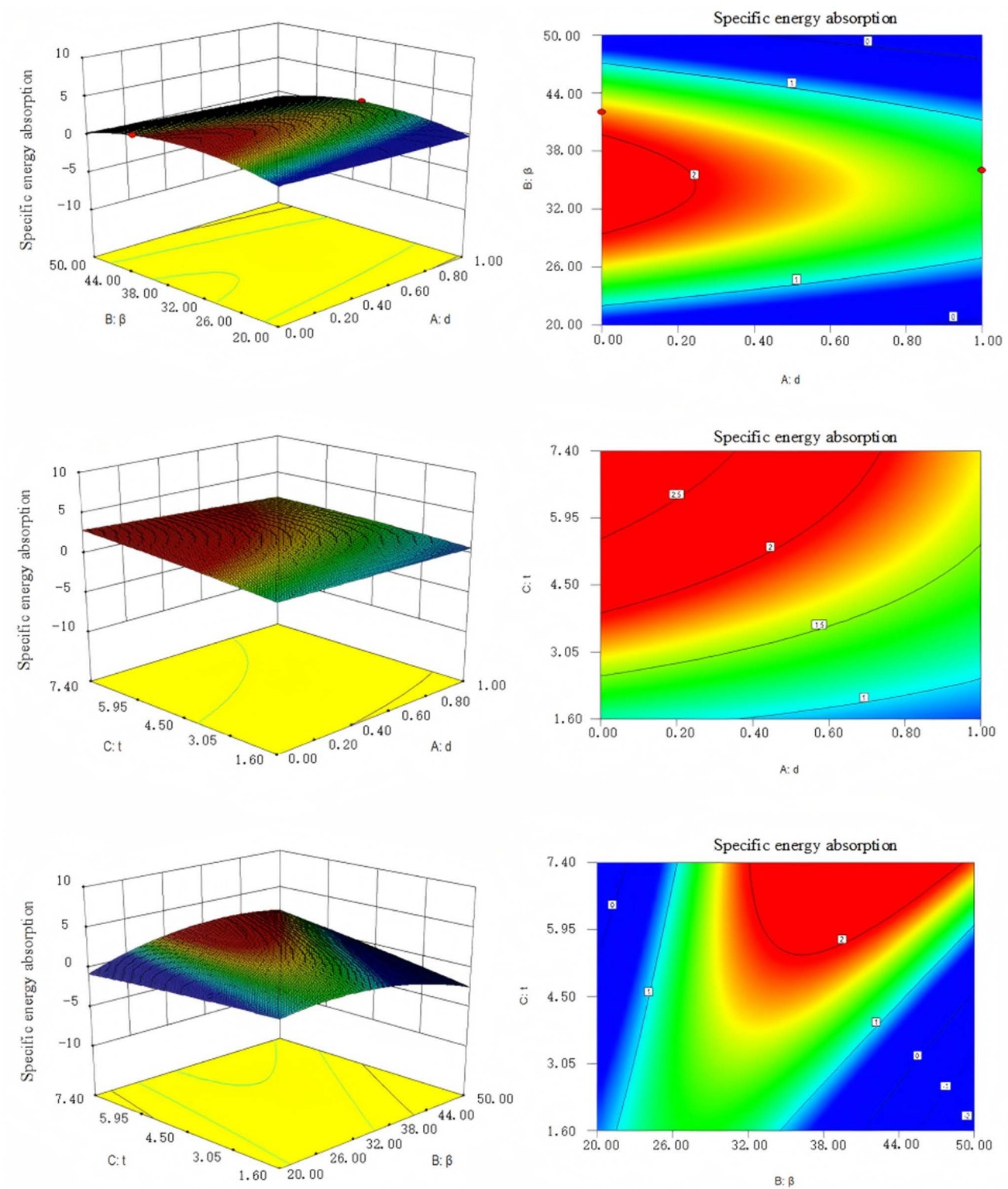

**Fig 7. The response surface of d, β and t on specific energy absorption.**

$$S_{EA}(d,\beta,t) = 1.79 - 0.41d - 0.16\beta + 0.68t - 0.055d\beta - 0.24dt + 1.44\beta t - 1.72\beta^2 - 0.16t^2 \tag{3.3}$$

$$R_E{}^2 = 0.9908 \tag{3.4}$$

$$R_{EA}{}^2 = 0.8992 \tag{3.5}$$

The R2 value close to 1 indicates a good fit of the equation and the test confirms that the regression of the equation is significant. The angle between the corrugated core plate of the hydraulic support energy absorber and the vertical direction (angle of inclination), as well as the thickness of the corrugated core, must fall within a certain dimensional range. The parameter model is as follows:

$$\begin{cases} \max(S_{EA}) \\ 20° \leq \beta \leq 50° \\ 1.8mm \leq t \leq 7.4mm \\ d \leq 1, d \in N \end{cases} \tag{3.6}$$

The Non-dominated Sorting Genetic Algorithm (NSGA-II) effectively addresses non-linear optimization problems and is widely used. Therefore, in this study, the NSGA-II algorithm is used for optimization, resulting in a Pareto optimal solution set. By comparing the evaluation results of the optimal solutions, the best configurations are determined as follows: the triangular corrugated unit has an optimal angle of 30° and a thickness of 2.9 mm; the trapezoidal corrugated unit has an optimal vertical angle of 34° and a thickness of 4.2 mm; and the sinusoidal corrugated unit has an optimal angle of 32° and a thickness of 3.2 mm. Among these, the sinusoidal structure is identified as the most optimal.

## 4. Simulation of corrugated energy absorber with three kinds of sandwich structures

In accordance with the optimized results of the corrugated core and the top beam parameters of the hydraulic support, this study has selected a portion of the total energy-absorbing component as the subject of the simulation. The dimensions of the simulation model are 300 mm × 300 mm × 60 mm. Three-dimensional models of the various corrugated components were created using SolidWorks and subsequently imported into the explicit dynamic module of ANSYS2022R2 Workbench. The parameters of component materials are shown in Tables 4–6. The corrugated structure model is discretized using a mesh. The corrugated cells can be considered as thin-shell structures, so shell elements are used for meshing the core. Face-to-face contact is applied between the rigid plates and components. During deformation, the parts of the corrugated cells are set for self-contact, with a friction coefficient of 0.25. Boundary conditions are applied to the model by fixing the lower supporting steel plate and applying a downward vertical force to the upper supporting steel plate, as shown in Fig 8.

The Johnson-Cook material model and failure model are commonly used to describe the strength limits and failure behavior of metallic materials under large deformation, high strain rates, and elevated temperatures. In the Johnson-Cook strength model, the yield stress is determined by strain, strain rate, and temperature.

Material Yield Model (Damage):

$$\sigma_Y = [A + B(\bar{\varepsilon}^{pl})^{n_1}]\left[1 + C\ln\left(\frac{\dot{\bar{\varepsilon}}^{pl}}{\dot{\varepsilon}_0}\right)\right](1 - T^{*m_n}) \tag{4.1}$$

$$T^* = \frac{T - T_{ref}}{T_{melt} - T_{ref}} \tag{4.2}$$

Where: $\sigma_Y$ is the equivalent flow stress; $\bar{\varepsilon}^{pl}$ is the equivalent plastic strain; $A$ is the initial yield stress; $B$ is the strain hardening modulus; $n_1$ is the strain hardening exponent; $B$ is the strain rate sensitivity coefficient; $m_n$ is the temperature sensitivity coefficient; $\dot{\varepsilon}_0$ is the reference strain rate; $T_{melt}$ is the melting temperature; $T_{ref}$ is the reference temperature.

**Table 4. Material parameters.**

| Materials | Modulus of elasticity (GPa) | Density(kg/m³) | Poisson's ratio | Yield strength (MPa) | Tensile strength (MPa) |
|---|---|---|---|---|---|
| Q235B | 210 | 7800 | 0.28 | 235 | 360 |

**Table 5. The Johnson-Cook constitutive model parameters of Q235B steel.**

| A(MPa) | B(MPa) | C | n1 | mn |
|---|---|---|---|---|
| 244.8 | 400.0 | 0.039 | 0.36 | 0.757 |

**Table 6. The Johnson-Cook failure criterion parameters of Q235B steel.**

| $d_1$ | $d_2$ | $d_3$ | $d_4$ | $d_5$ |
|---|---|---|---|---|
| -43.408 | 44.608 | -0.016 | 0.0145 | 0.046 |

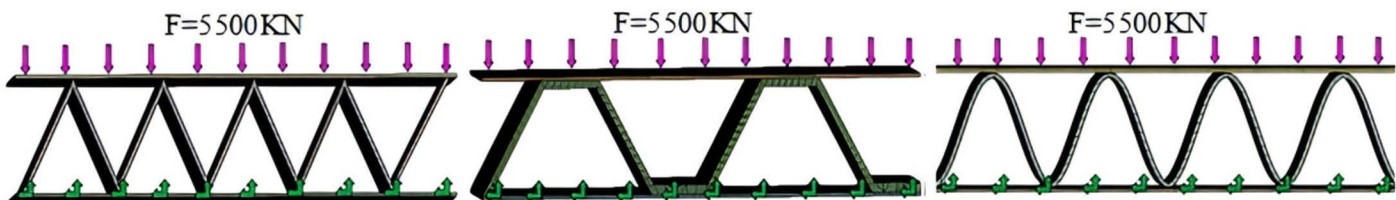

**Fig 8. Loading of mesh boundary conditions for 3D models of different corrugated structures.**

Material Failure Model:

$$\varepsilon_f = [d_1 + d_2 \exp(-d_3\eta)] \left[1 + d_4 \ln\left(\frac{\dot{\varepsilon}^{pl}}{\dot{\varepsilon}_0}\right)\right] (1 + d_5 T^*)$$

(4.3)

Where: d1、d2、d3、d4、d5 are material failure parameters, and $\eta$ represents the stress triaxiality.

A three-dimensional model of a triangular corrugated structure was subjected to conditional loading. The base plate was fixed in place, and a velocity of 5 m/s was applied to the top plate. The lateral corrugations were constrained, thus allowing only vertical motion. The analysis yielded results pertaining to the energy absorption, support reaction forces, and buckling deformation of the single-layer triangular corrugated structure which are shown in Fig 9–11–.

As illustrated in Fig 9, the force-displacement curve demonstrates that following a displacement of 50 mm, the rate of increase in force rises markedly, indicating that the energy-absorbing component enters a compression phase. In the 0-8 mm range, the force increases almost linearly, indicating that this stage corresponds to the linear elastic phase of the energy-absorbing component. In the interval between 8 and 50 mm, the support reaction force remains relatively stable with minimal variation, indicating the plastic deformation phase. Overall, the alterations in the support reaction force are gradual and insignificant, indicating that the single-layer triangular corrugated core is an optimal choice for an energy-absorbing component.

During the meshing of the sinusoidal corrugated structure, a thickness error of up to 0.01 mm was observed. The edges of the corrugated component were fixed, and the remaining loading conditions were maintained at the same levels as those applied to the triangular structure.

An analysis of Figs 12, 13 and the buckling deformation shown in Fig 14 indicates that the force-displacement curve of the sinusoidal corrugated core structure is almost linear between 0-3 mm, reaching a peak crush load at approximately

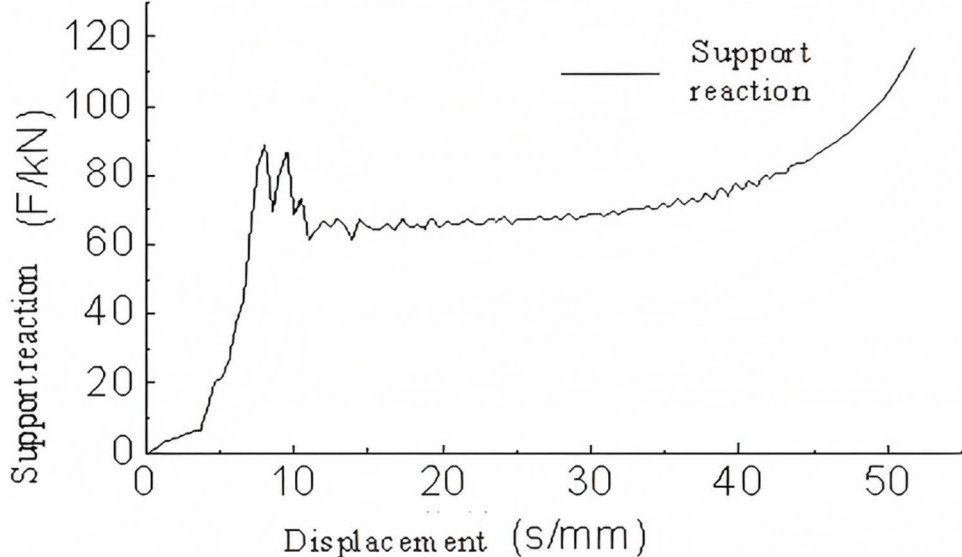

**Fig 9. Support reaction-displacement curve of single-layer triangular corrugated.**

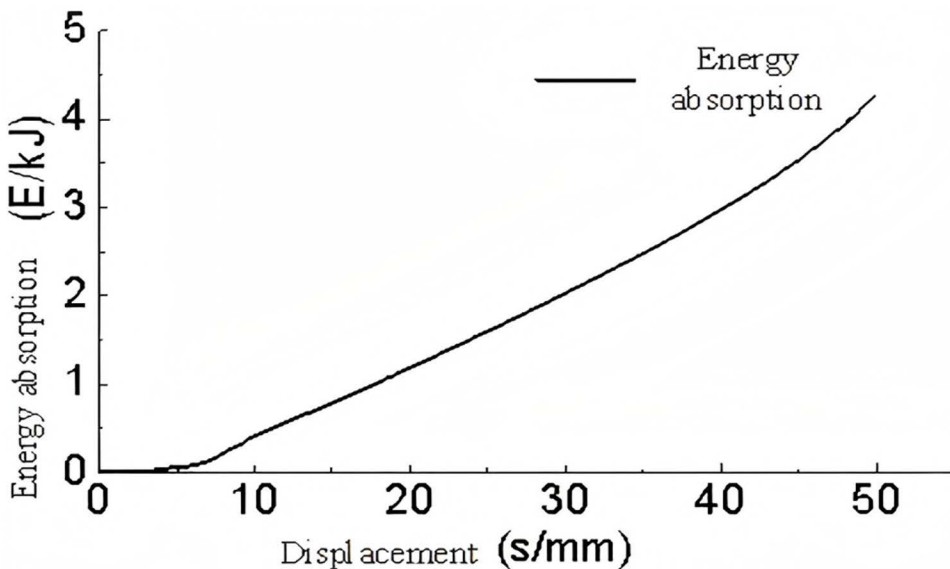

**Fig 10. Energy absorption-displacement curve of single layer triangular corrugated members.**

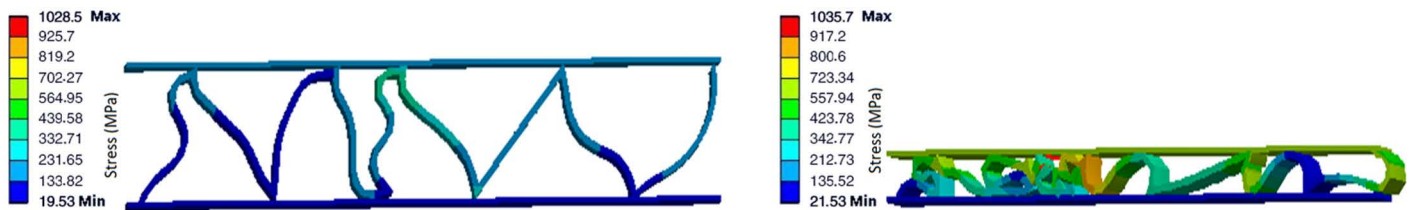

**Fig 11. Buckling deformation of single layer triangular corrugated members.**

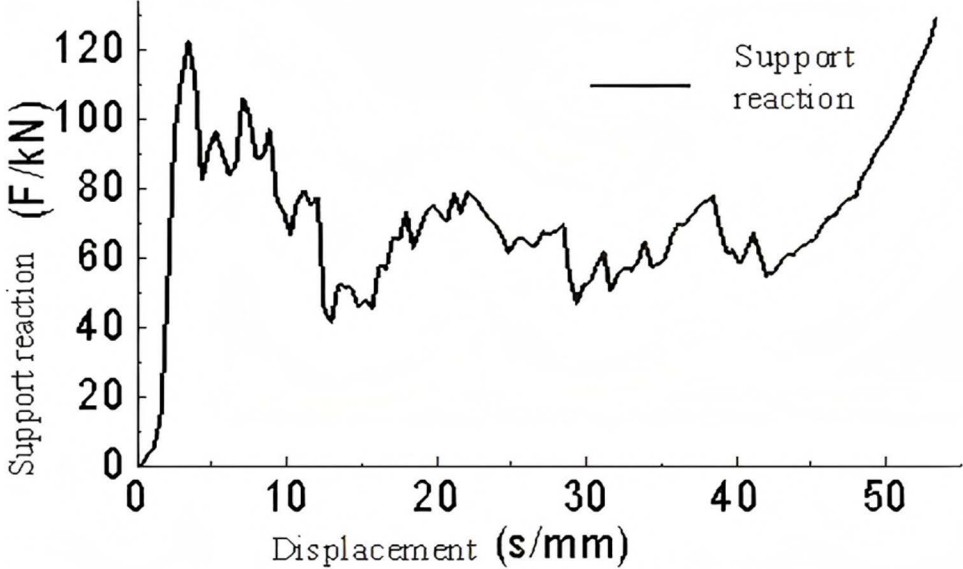

**Fig 12. Support reaction-displacement curve of single-layer sinusoidal corrugated members.**

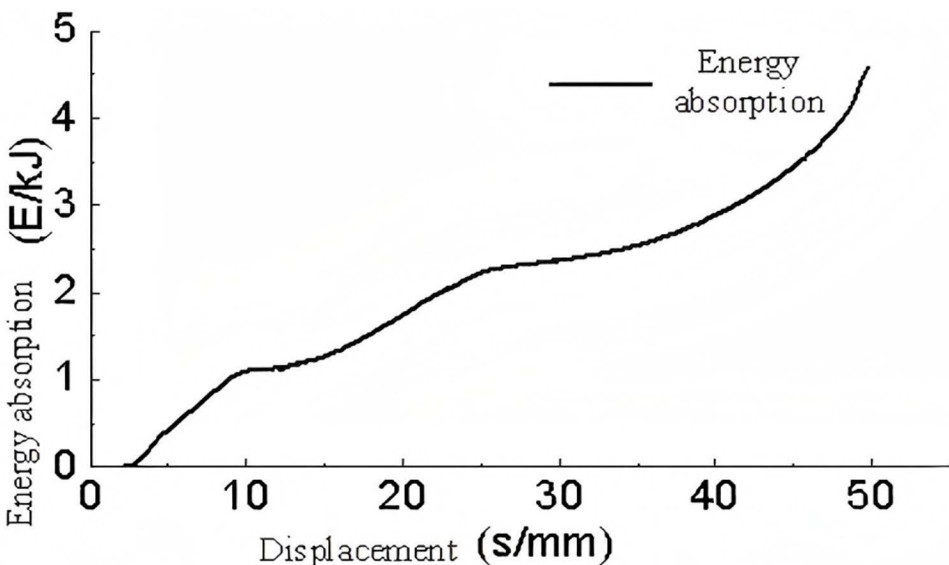

**Fig 13. Energy absorption-displacement curve of single-layer sinusoidal corrugated members.**

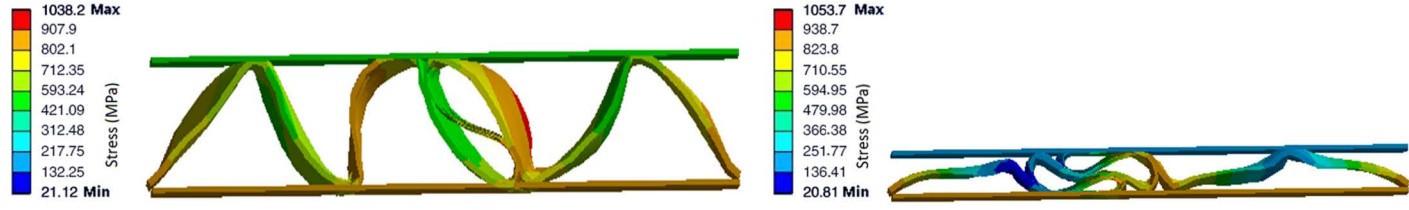

**Fig 14. Buckling deformation of single-layer sinusoidal corrugated members.**

3 mm. This corresponds to the linear elastic phase. In the range of 3-50 mm, the variations in force are considerable but remain within 1.3 times the peak crush load, indicating the presence of plastic deformation within the effective deformation displacement range. This phase represents the maximum energy absorption capacity. Subsequently, beyond a displacement of 50 mm, the rate of force increase becomes exponential, accompanied by a notable rise in the energy absorption rate, which indicates the onset of the compression phase. The buckling deformation and the force-displacement curve indicate that when the corrugations make contact, the force increases. Conversely, when the contact force dissipates, the force decreases accordingly.

During the loading process of the trapezoidal corrugated component, the incomplete upper base of the trapezoidal corrugation necessitates the limitation of displacement of the lower edge in the x and y directions. This guarantees that the component will remain in contact with the upper and lower steel plates. The remaining loading conditions are identical to those applied to the triangular corrugated member. The analysis provides the support reaction forces, energy absorption and buckling deformation behavior of the trapezoidal corrugated structure which are shown in Figs 15–17–.

As illustrated in Fig 15, the trapezoidal corrugated core structure exhibits a relatively rapid rate of force increase within the 0-10 mm range. This phase can be considered to be in the linear elastic regime, given that the displacement is greater than would be expected from a purely linear elastic behavior, particularly when combined with the buckling deformation phase. In the 12-45 mm range, the rate of force change is relatively stable and can be considered to represent the plastic deformation phase. Subsequently, at a point of 45 mm, the rate of force increase becomes exponential, indicating the onset of the compression phase.Compared to the other two structures, the trapezoidal corrugated core exhibits more intense fluctuations, both in magnitude and frequency. These fluctuations are associated with localized yielding, crack propagation, and non-uniform deformation, indicating that the material has reached its critical failure point at this stage.

The results of the single-layer analysis of the three different corrugated core structures were calculated using the relevant theoretical formulae, and the resulting values are presented in Table 7.

As demonstrated in Table 7, the simulation outcomes indicate that the sinusoidal corrugated core structure exhibits the highest energy absorption, followed by the trapezoidal corrugated core, with the triangular corrugated core demonstrating the lowest. With regard to specific energy absorption, the sinusoidal corrugated core exhibits the highest value, followed by the triangular and then the trapezoidal. Furthermore, the sinusoidal corrugated core exhibits the highest peak load, load fluctuation coefficient and peak load reduction coefficient, indicating that its support reaction force curve

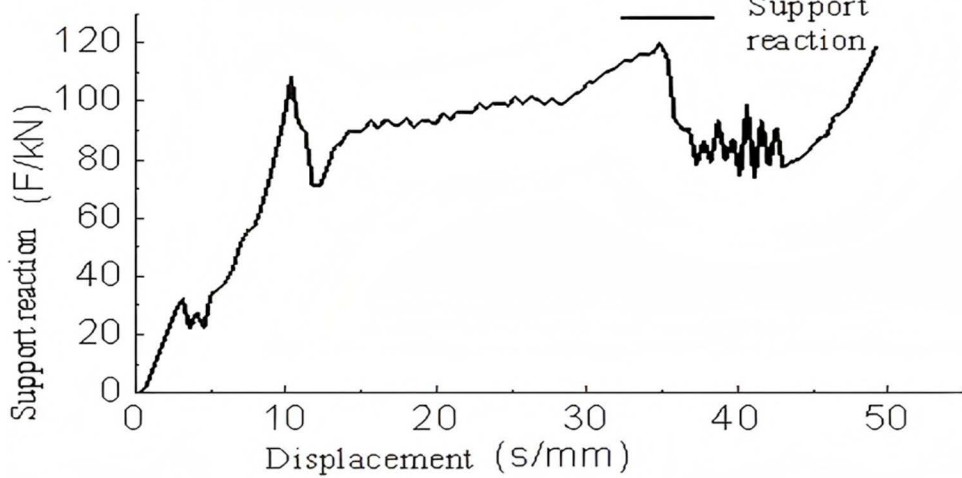

**Fig 15. Support reaction-displacement curve of single-layer trapezoidal corrugated members.**

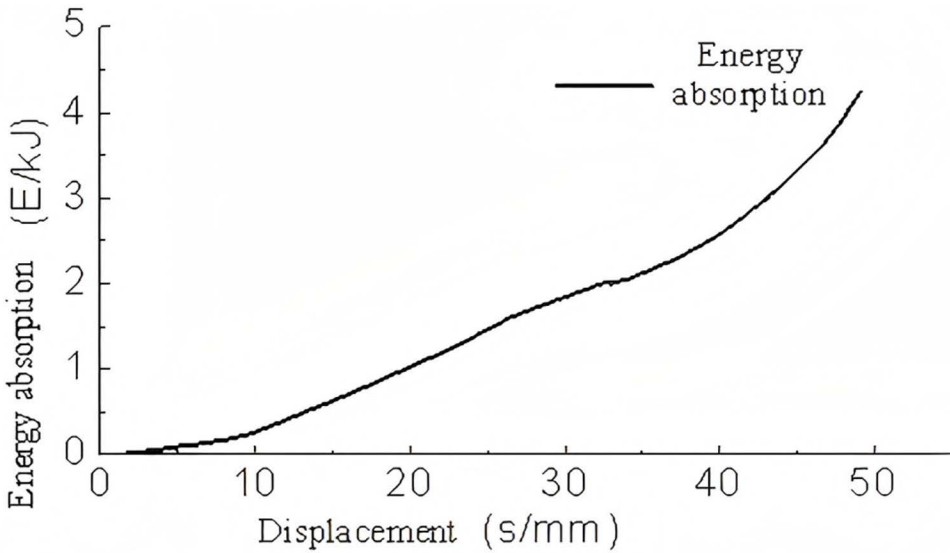

**Fig 16. Energy absorption-displacement curve of single ladder corrugated members.**

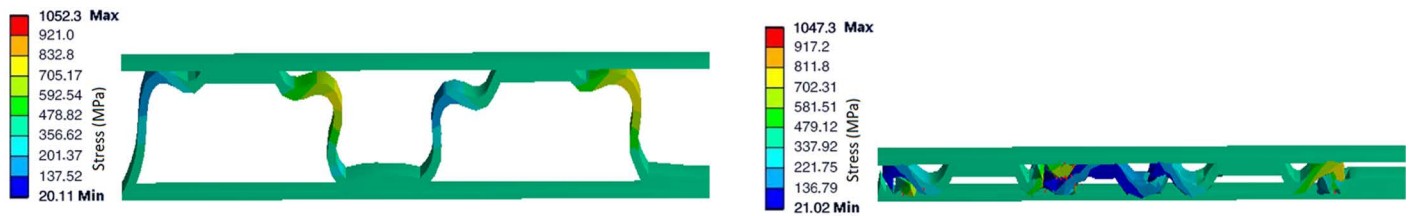

**Fig 17. Buckling deformation of single ladder corrugated members.**

**Table 7. Evaluation index of single layer various corrugated components.**

| Evaluation indicators | Total energy absorption $E_z$/kJ | Specific energy absorption $E_{SEA}$/kJ/kg | Peak Crush Load $F_{max}$/kN | Load fluctuation factor $\triangle$ | Giving way capacity factor $\gamma_s$ | Peak load reduction factor $\xi$ |
|---|---|---|---|---|---|---|
| Triangular corrugated core structure | 4.25 | 1.81 | 91 | 1.12 | 0.80 | 0.32 |
| Sinusoidal corrugated core structure | 4.86 | 1.85 | 124 | 1.25 | 0.83 | 1.48 |
| Trapezoidal corrugated core structure | 4.42 | 1.46 | 108 | 1.20 | 0.73 | 0.41 |

demonstrates the greatest fluctuation with displacement. The triangular corrugated core exhibits the lowest peak load, load fluctuation coefficient and peak load reduction coefficient, resulting in the most stable support reaction force curve. The trapezoidal corrugated core exhibits a stability that is intermediate to that of the other two. In consideration of the aforementioned performance indicators, it can be concluded that the top beam energy-absorbing component is required to demonstrate relatively high energy absorption and displacement capacity. Among the single-layer corrugated energy-absorbing components, the sinusoidal corrugated core structure is the most optimal in terms of the aforementioned performance indicators.

## 5. Experimental study on energy absorbing components of corrugated layer

The designed corrugated components are fabricated and tested to analyze the impact of different structures on energy absorption characteristics and verify their feasibility as energy-absorbing components. A comparative analysis of the experimental results is conducted, and the optimal energy-absorbing structure is selected based on the comprehensive consideration of various performance indicators.

### 5.1. Processing and preparation of components

To match the simulation, the test specimens were fabricated similarly to the simulation models using Q235B material with dimensions of 300×300×60mm.The edges of the corrugated cores were welded to the top and bottom steel plates. Within the triangular structures, only the contact areas between the corrugated core and the steel plates were welded, with a weld length of approximately one tenth of the contact length, primarily to secure the core. For the trapezoidal corrugated core structure, due to the flatness of the top and bottom, the internal contacts of the core were not welded.

Only the contact points at the edges of the core were welded. As shown in Fig 18, the fabricated single layer specimens have an allowable height error range of -2 to 2mm.

### 5.2. Test purpose and procedure

The purpose of the experiment is to examine the force-displacement deformation behavior of the corrugated energy-absorbing component with three core structures at a single-layer height, while also evaluating its feasibility as an energy-absorbing component. The experiment utilizes a YAW-2000 laboratory hydraulic press and leveling device, as shown in Figs 19 and 20. The press is connected to a computer and a force-displacement data acquisition system (with a relative error in the actual force of less than ±1% of the indicated value) to conduct quasi-static compression tests on the components, with a compression rate set to 0.5mm/s. To ensure the press platform maintains a vertical pressure direction during compression, a leveling device is used to adjust the platform before placing the energy-absorbing component.

The experimental procedure is as follows

1. Check the experimental equipment to ensure that it is intact and the operating platform is safe, then turn on the power.

2. Adjust and level the press, ensuring that the upper and lower contact surfaces are as horizontal as possible.

3. Adjust the upper platen of the press to make contact with the specimen.

4. Activate the data acquisition system, select the loading method, set the loading speed, reset other parameters to zero and start the press to perform the static compression test.

5. When compression is complete, unload the press, adjust the top platen to disengage from the specimen, remove the specimen, save the test data, and turn off the power.

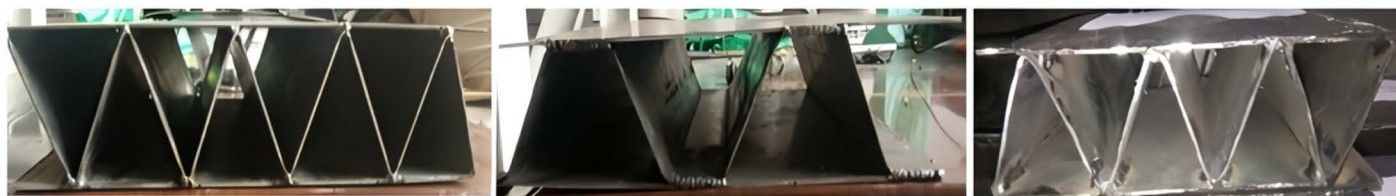

**Fig 18. Processing objects of three kinds of corrugated core components.**

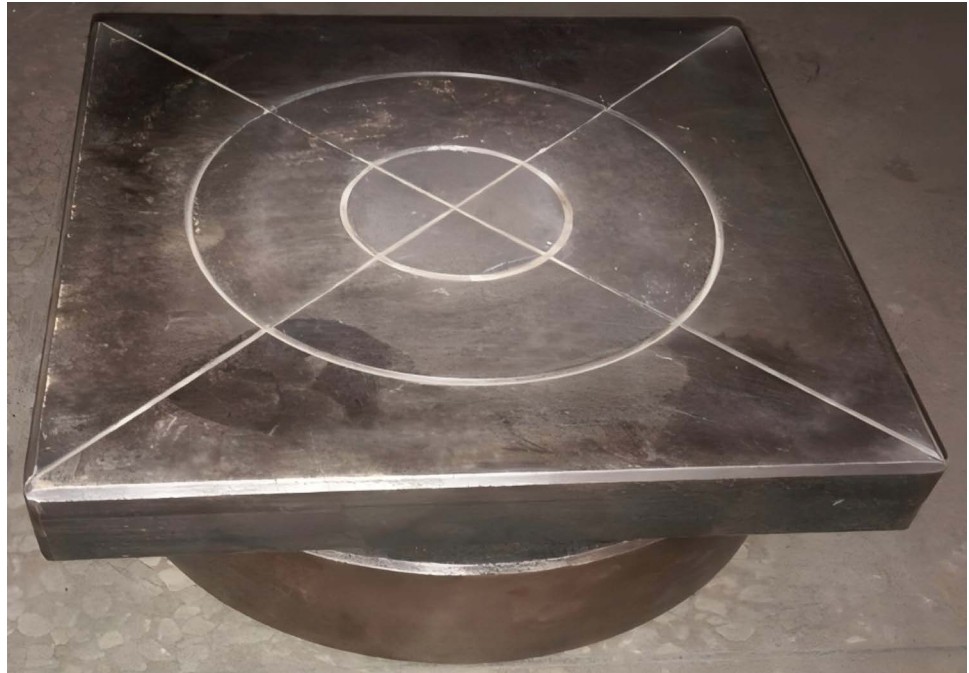

**Fig 19. Leveling device.**

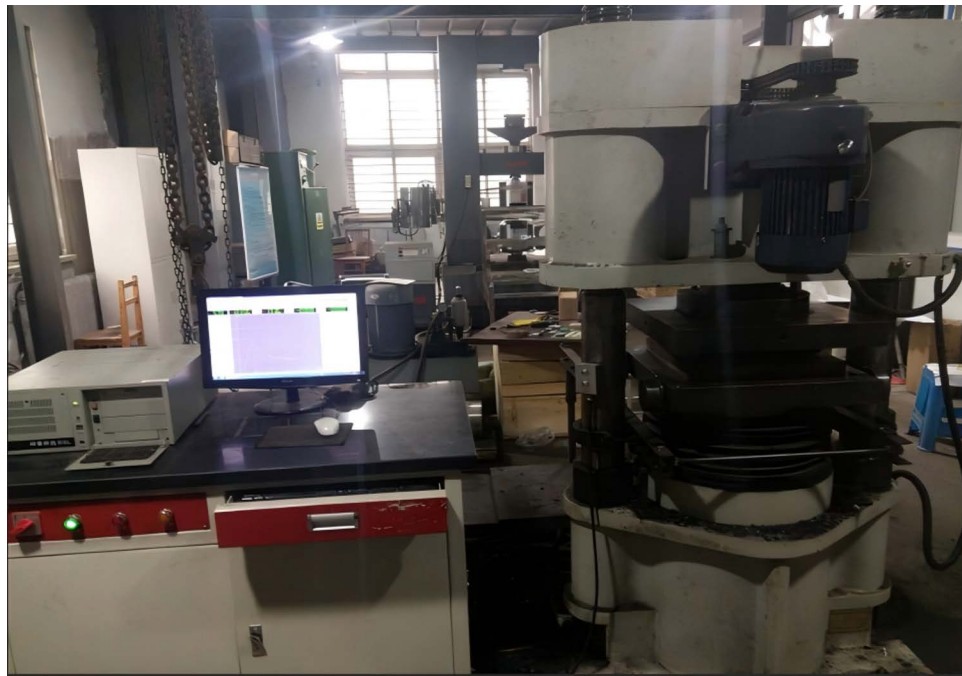

**Fig 20. YAW-2000 hydraulic dynamometer.**

## 5.3. Test results and analysis

Fig 21 presents the force-displacement data curves for different corrugated core components.

From the analysis of the support force-displacement curve in Fig 21, it can be seen that the trapezoidal corrugated core structure reaches the peak crushing load first, followed by the triangular and then the sinusoidal shape. Consequently, in the linear elastic stage, the trapezoidal structure has the smallest displacement, while the sinusoidal has the largest.

During the plastic deformation phase, the triangular structure first reaches the densification phase with a relatively small displacement of about 12-50 mm. This is followed by the sinusoidal shape with a plastic deformation displacement of approximately 13-56 mm. The trapezoidal corrugated core structure shows the largest displacement in the plastic deformation stage, ranging from approximately 7-53 mm.

Observing the compression deformation process, both the triangular and sinusoidal cores tend to contact the top and bottom plates early in the deformation. As a result, the support force continues to increase after the peak compression load has dropped to a certain value. Comparing the two, the sinusoidal shape has a higher support force during the plastic deformation phase, while the plastic deformation and compression phases of the triangular shape are less distinct.

The trapezoidal corrugated core structure, with larger gaps between the core plates, does not experience overlap or contact during deformation. Therefore, it remains relatively stable during the plastic deformation phase compared to the other two shapes.

Overall, when comparing the three structures based on the energy absorption criteria, Table 8 provides a comparison of the data for each indicator.

As shown in Table 8, among the single-layer components, the sinusoidal corrugated core has the highest energy absorption, specific energy absorption and displacement capacity coefficient. The triangular corrugated core structure has the lowest energy absorption, while the trapezoidal corrugated core structure has the lowest specific energy absorption. The sinusoidal corrugated core also has the lowest load fluctuation coefficient and peak load reduction coefficient, making it the most effective in cushioning energy absorption and displacement among the single layer structures.

Figs 22–24 compare the results of simulation and compression deformation experiments. Table 9 presents a comparison between the experimental and simulation results for the energy absorption indicators of corrugated components.

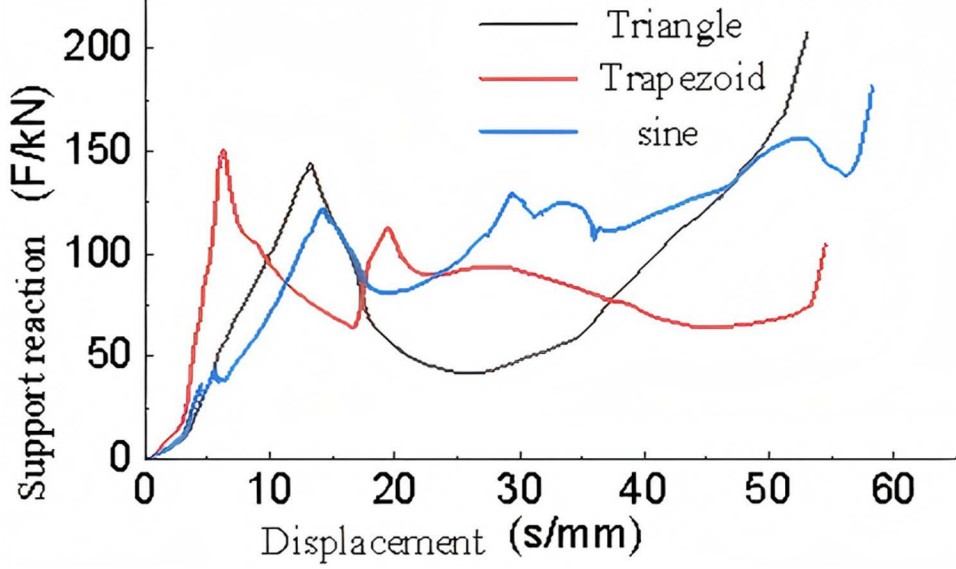

**Fig 21. Supporting reaction-displacement curves of single-layer structures with different corrugated cores.**

**Table 8. Evaluation index of single layer various corrugated components.**

| Evaluation indicators | Total energy absorption $E_z$/kJ | Specific energy absorption $E_{SEA}$/kJ/kg | Peak Crush Load $F_{max}$/kN | Load fluctuation factor $\Delta$ | Giving way capacity factor $\gamma_s$ | Peak load reduction factor $\xi$ |
|---|---|---|---|---|---|---|
| Triangular corrugated core structure | 3.99 | 1.73 | 144 | 1.92 | 0.77 | 1.12 |
| Sinusoidal corrugated core structure | 4.65 | 1.77 | 122 | 1.67 | 0.88 | 0.49 |
| Trapezoidal corrugated core structure | 4.32 | 1.42 | 150 | 1.89 | 0.82 | 1.30 |

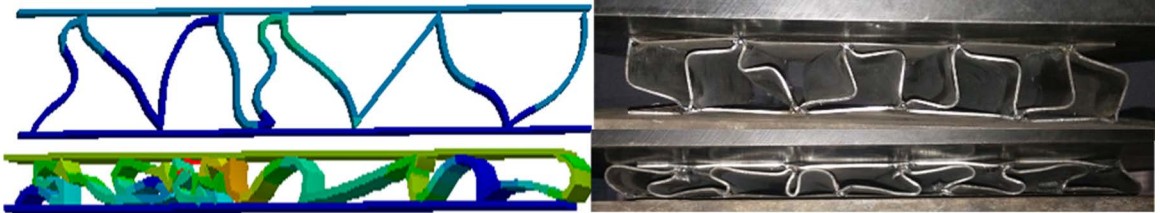

**Fig 22. Comparison between simulation and experiment of triangular corrugated member.**

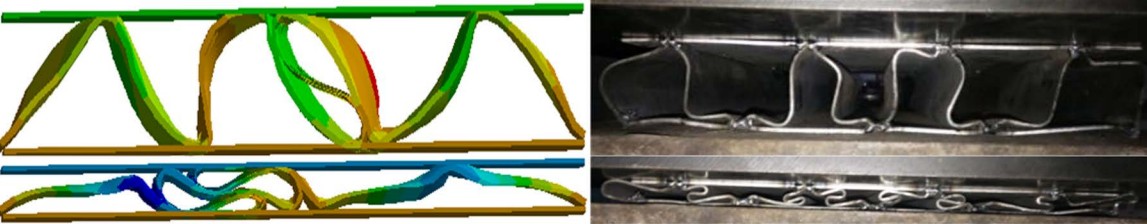

**Fig 23. Comparison between simulation and experiment of sinusoidal corrugated components.**

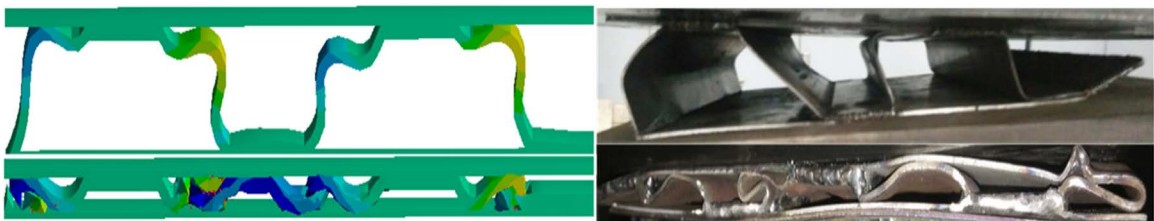

**Fig 24. Comparison between simulation and experiment of trapezoidal corrugated member.**

From the analysis of Figs 22, 23, 24, and the corresponding table, the total relative deviations for total energy absorption, specific energy absorption, and load fluctuation coefficient are 7%, 5%, 42%, 5%, 5%, 25%, 2%, 3%, and 37%, respectively. Among these, the load fluctuation coefficient exhibits a significant deviation, primarily due to local thickness variations caused by heat pressing during the manufacturing process of the energy-absorbing components. Additionally, residual stresses were introduced during the welding process, and since no heat treatment was performed to relieve them, the stress distribution in the specimen became non-uniform under compressive loading during testing. Overall, the experimental and simulation results exhibit good agreement.

**Table 9. Comparison between experimental and simulation results of some energy absorption indexes.**

| Evaluation indicators | Total energy absorption $E_z$/kJ | Specific energy absorption $E_{SEA}$/kJ/kg | Load fluctuation factor $\Delta$ |
|---|---|---|---|
| Triangular corrugated core structure simulation result | 4.25 | 1.81 | 1.12 |
| Triangular corrugated core structure test value | 3.99 | 1.73 | 1.92 |
| Relative deviation/% | -7 | -5 | 42 |
| Sinusoidal corrugated core structure simulation result | 4.86 | 1.85 | 1.25 |
| Sinusoidal corrugated core structure test value | 4.65 | 1.77 | 1.67 |
| Relative deviation/% | -5 | -5 | 25 |
| Trapezoidal corrugated core structure simulation result | 4.42 | 1.46 | 1.20 |
| Trapezoidal corrugated core structure test value | 4.32 | 1.42 | 1.89 |
| Relative deviation/% | -2 | -3 | 37 |

## Conclusions

(1)  A corrugated energy-absorbing component with three different core structures is proposed for application in the top beam of a hydraulic support. Based on a mechanical model, the parametric equations describing the relationships among different variables are analyzed, providing theoretical support for simulations and experiments.

(2) For the proposed corrugated structures, the parameters d, β and t were selected at a certain height of the single layer corrugated structure. A mathematical model was developed and solved using the response surface methodology for optimization. The optimized results for energy absorption and specific energy absorption showed that the optimum angle for the triangular corrugated unit was 30° with a thickness of 2.9 mm, for the trapezoidal corrugated unit was a vertical angle of 34° with a thickness of 4.2 mm, and for the sinusoidal corrugated unit was 32° with a thickness of 3.2 mm. The sinusoidal structure was found to be the most optimal.

(3) Experimental studies were conducted on the specimens and compared with simulation results, showing total relative deviations of 7%, 5%, 42%, 5%, 5%, 25%, 2%, 3%, and 37% for total energy absorption, specific energy absorption, and load fluctuation coefficient, respectively. Overall, the experimental and simulation results exhibit good agreement, confirming the high reliability of the theoretical model. Compared to the other two structures, the sinusoidal corrugated core exhibits the highest energy absorption, specific energy absorption, and accommodation capacity coefficient, while achieving the lowest load fluctuation coefficient and peak load reduction coefficient. As a result, it demonstrates the most effective energy absorption and displacement accommodation performance.

(4) Compared to previous studies, the innovation of this paper lies in applying corrugated structures to energy-absorbing hydraulic supports for collision protection. Additionally, an optimization strategy combining response surface methodology and single-factor experimental methods is used, along with a validation approach integrating theory and experimentation. This provides new insights and technical support for enhancing the energy absorption capacity of hydraulic support systems.

## Author contributions

**Writing – original draft:** Qian Liu, Jiyang Meng, Peng Yang, Xisheng Yu.

**Writing – review & editing:** Zuen Shang.

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
