## [Decision Letter · Decision Letter 0]

5 Feb 2025

PONE-D-24-57458Optimization design of top beam energy absorbing member of corrugated hydraulic support with three kinds of sandwich structures based on adaptive response surface methodPLOS ONE

Dear Dr. Liu,

Thank you for submitting your manuscript to PLOS ONE. After careful consideration, we feel that it has merit but does not fully meet PLOS ONE’s publication criteria as it currently stands. Therefore, we invite you to submit a revised version of the manuscript that addresses the points raised during the review process.

We look forward to receiving your revised manuscript.

Kind regards,

Yuan-Fong Chou Chau

Academic Editor

PLOS ONE

Journal Requirements:

“Tis study was funded by National Natural Science Foundation of China (Youth) Project: Dynamic 3D Scene Perception of Coal Mine Catastrophic Search and Rescue Robot Based on Multimodal Information Fusion; Project No.: 5240042310 (2024, under research) Liaoning Provincial Department of Education Youth Improvement Project: 3D Environmental Perception of Coal Mine Disaster Scene Inspection Robot; (Chaired in 2024, under research) Liaoning Provincial Department of Science and Technology Project: Research on Posture Control and Visual Cognition Algorithm of Coal Mine Rescue Robot Based on Origami Theory; Project No.: 2023-BS-204 (Principal Investigator in 2023). “

“Tis study was funded by National Natural Science Foundation of China (Youth) Project: Dynamic 3D Scene Perception of Coal Mine Catastrophic Search and Rescue Robot Based on Multimodal Information Fusion; Project No.: 5240042310 (2024, under research) Liaoning Provincial Department of Education Youth Improvement Project: 3D Environmental Perception of Coal Mine Disaster Scene Inspection Robot; (Chaired in 2024, under research) Liaoning Provincial Department of Science and Technology Project: Research on Posture Control and Visual Cognition Algorithm of Coal Mine Rescue Robot Based on Origami Theory; Project No.: 2023-BS-204 (Principal Investigator in 2023). “

“The author(s) received no specific funding for this work”

6. We note that your Data Availability Statement is currently as follows: All relevant data are within the manuscript and its Supporting Information files.

**Additional Editor Comments:**

Comments from PLOS Editorial Office: We note that one or more reviewers has recommended that you cite specific previously published works. As always, we recommend that you please review and evaluate the requested works to determine whether they are relevant and should be cited. It is not a requirement to cite these works. We appreciate your attention to this request.

Reviewers' comments:

Reviewer's Responses to Questions

**Comments to the Author**

1. Is the manuscript technically sound, and do the data support the conclusions?

Reviewer #1: Yes

Reviewer #2: Partly

2. Has the statistical analysis been performed appropriately and rigorously? 

Reviewer #1: Yes

Reviewer #2: N/A

3. Have the authors made all data underlying the findings in their manuscript fully available?

Reviewer #1: Yes

Reviewer #2: Yes

4. Is the manuscript presented in an intelligible fashion and written in standard English?

Reviewer #1: Yes

Reviewer #2: Yes

5. Review Comments to the Author

Reviewer #1: Overview and general recommendations:

This is an article about the application of corrugated structure in impact absorbing advanced support hydraulic pillar. According to the national design standard for advanced support hydraulic pillars, the corrugated energy-absorbing components of the support beams are optimised and their cushioning and energy-absorbing characteristics are investigated. The manuscript is interesting and the reviewer recommends acceptance with modifications. Specifically, the following points should be noted for improvement:

1. The expression of several proper nouns in the manuscript is inconsistent, so please check it carefully. For example, the expressions related to the three energy-absorbing components are inconsistent.

2. In subsection 1.2, it is mentioned that ‘When the reaction force exceeds the initial peak crushing load and after hydraulic support safety valve failures, the deformation displacement is considered effective if the reaction force is less than 1.3 times the initial peak crushing load.’ What is the basis for this, please include necessary explanations or references.

3.The introductory part deals with only about 20% of the references in the last three years, which should be increased appropriately to enrich the study. (Recommended as follows: https://doi.org/10.1016/j.tws.2024.112415

https://doi.org/10.1007/s43452-024-00895-9

https://doi.org/10.1016/j.engfracmech.2024.110009)

4. Figures 3.2 and 3.3 seem to lack the necessary textual descriptions in the manuscript; moreover, the titles of the figures are inconsistent with the contents of the figures (e.g., energy absorption vs. specific energy absorption in Fig. 3.2; relative energy absorption vs. specific energy absorption in Fig. 3.2).

5. The curves in Figure 4.8 are interesting and, unlike the other two structures, the reader may expect a more in-depth analysis.

Reviewer #2: The paper has produced some meaningful results, but there are still many issues that need to be revised.

1. The structural layout is not good enough and the readability is poor.

2. Innovation needs to be emphasized. Is it just a matter of different application scenarios? Is there any innovation in the corrugated structure itself?

3.The ANSYS version should be provided. I didn't see the introduction and steps of calculation by using ANSYS in the article, please provide detailed information.

4.What is the purpose of the experiment listed in the article, and is it to verify the calculation results? I did not see the verification process and results. If it's not for verification, is it using experiments as a means of research? So what is the connection with the previous calculation, and what is the logic behind putting it into a paper?

6. PLOS authors have the option to publish the peer review history of their article (what does this mean? ). If published, this will include your full peer review and any attached files.

**Do you want your identity to be public for this peer review?** For information about this choice, including consent withdrawal, please see our Privacy Policy .

Reviewer #1: No

Reviewer #2: No

---

## [Author Response · Author response to Decision Letter 1]

26 Mar 2025

Dear editor

Thank you for your letter and for the reviewers’ comments concerning our manuscript entitled “Optimization design of top beam energy absorbing member of corrugated hydraulic support with three kinds of sandwich structures based on adaptive response surface method” (PONE-D-24-57458). Those comments are very valuable and very helpful for revising and improving our paper, as well as the important guiding significance to our researches. We have studied comments carefully and have made correction which we hope meet with approval. In the revision, we have fully addressed the comments made by the reviewers and the editor. The completed changes are as follows:

Reviewer #Editor:

Comment 1

and https://journals.plos.org/plosone/s/file?id=ba62/PLOSOne_formatting_sample_title_authors_affiliations.pdf

Response 1

The full text format has been revised and checked in plos one style, and I would appreciate your approval.

Comment2

Please note that PLOS ONE has specific guidelines on code sharing for submissions in which author-generated code underpins the findings in the manuscript. In these cases, we expect all author-generated code to be made available without restrictions upon publication of the work.

Please review our guidelines at https://journals.plos.org/plosone/s/materials-and-software-sharing#loc-sharing-code and ensure that your code is shared in a way that follows best practice and facilitates reproducibility and reuse.

Response2

Ok, thanks for the suggestion. We have submitted the code for the article to the public repository

https://github.com/001001-png/001/commit/326900785861f2fc65de8878a292727bce316d91

Comment3

We note that the grant information you provided in the ‘Funding Information’ and ‘Financial Disclosure’ sections do not match.

Response3

This issue has been corrected, thank you for your feedback .

Comment 4

 Thank you for stating the following financial disclosure:

“Tis study was funded by National Natural Science Foundation of China (Youth) Project: Dynamic 3D Scene Perception of Coal Mine Catastrophic Search and Rescue Robot Based on Multimodal Information Fusion; Project No.: 5240042310 (2024, under research) Liaoning Provincial Department of Education Youth Improvement Project: 3D Environmental Perception of Coal Mine Disaster Scene Inspection Robot; (Chaired in 2024, under research) Liaoning Provincial Department of Science and Technology Project: Research on Posture Control and Visual Cognition Algorithm of Coal Mine Rescue Robot Based on Origami Theory; Project No.: 2023-BS-204 (Principal Investigator in 2023). “

Response 4

This issue has been corrected, thank you for your feedback .

Comment 5

Thank you for stating the following in the Acknowledgments Section of your manuscript:

“Tis study was funded by National Natural Science Foundation of China (Youth) Project: Dynamic 3D Scene Perception of Coal Mine Catastrophic Search and Rescue Robot Based on Multimodal Information Fusion; Project No.: 5240042310 (2024, under research) Liaoning Provincial Department of Education Youth Improvement Project: 3D Environmental Perception of Coal Mine Disaster Scene Inspection Robot; (Chaired in 2024, under research) Liaoning Provincial Department of Science and Technology Project: Research on Posture Control and Visual Cognition Algorithm of Coal Mine Rescue Robot Based on Origami Theory; Project No.: 2023-BS-204 (Principal Investigator in 2023). “

“The author(s) received no specific funding for this work”

Response 5

This issue has been corrected, thank you for your feedback .

Comment 6

We note that your Data Availability Statement is currently as follows:

"All relevant data are within the manuscript and its Supporting Information files."

Response 6

A change has been made in the submission system to address this issue.

Reviewer #1: 

Comment 1

The expression of several proper nouns in the manuscript is inconsistent, so please check it carefully. For example, the expressions related to the three energy-absorbing components are inconsistent.

Response 1

Thank you for your valuable feedback. We appreciate your careful review and will thoroughly check the manuscript to ensure consistency in the expression of proper nouns. We will specifically review and standardize the terminology related to the three energy-absorbing components to maintain clarity and coherence. Please let us know if there are any additional areas you would like us to refine.

Comment 2

In subsection 1.2, it is mentioned that ‘When the reaction force exceeds the initial peak crushing load and after hydraulic support safety valve failures, the deformation displacement is considered effective if the reaction force is less than 1.3 times the initial peak crushing load.’ What is the basis for this, please include necessary explanations or references.

Response 2

Thank you for your insightful feedback. This standard is mainly based on the theory of material mechanics and the safety standards of engineering design. When the reaction force exceeds the initial peak crushing load, 1.3 times the reaction force can still maintain the effective deformation displacement of the structure, which is based on the plastic deformation capacity of the material and the safety margin of the design. Therefore, such a standard is usually used in engineering practice to ensure that the structure can still work effectively after damage or yield, and to ensure a certain degree of safety.

Comment 3

The introductory part deals with only about 20% of the references in the last three years, which should be increased appropriately to enrich the study. (Recommended as follows: https://doi.org/10.1016/j.tws.2024.112415
https://doi.org/10.1007/s43452-024-00895-9

https://doi.org/10.1016/j.engfracmech.2024.110009)

Response 3

Thank you for your valuable feedback. We appreciate your suggestion and recognize the importance of incorporating more recent references to strengthen the study. We will appropriately increase the proportion of references from the last three years to ensure the introductory section reflects the latest developments in the field. Additionally, we will carefully review the recommended articles and integrate relevant insights to further enrich the study's background and justification.

Please let us know if you have any specific areas where additional references would be particularly beneficial.

Comment 4

Figures 3.2 and 3.3 seem to lack the necessary textual descriptions in the manuscript; moreover, the titles of the figures are inconsistent with the contents of the figures (e.g., energy absorption vs. specific energy absorption in Fig. 3.2; relative energy absorption vs. specific energy absorption in Fig. 3.2).

Response 4

Thank you for your valuable feedback. We will carefully review Figures 3.2 and 3.3 to ensure that their textual descriptions are comprehensive and clearly aligned with the manuscript content. Additionally, we will verify and correct any inconsistencies between the figure titles and their actual content (e.g., distinguishing between energy absorption, specific energy absorption, and relative energy absorption in Figure 3.2). These revisions will help enhance the clarity and accuracy of the presented results.

We appreciate your attention to detail and will make the necessary improvements accordingly. Please let us know if you have any specific recommendations regarding the preferred terminology.

Comment 5

The curves in Figure 4.8 are interesting and, unlike the other two structures, the reader may expect a more in-depth analysis.

Response 5

Thank you for your insightful feedback. We appreciate your observation regarding the uniqueness of the curves in Figure 4.8 and understand the need for a more in-depth analysis. We will expand our discussion to provide a detailed interpretation of the underlying mechanisms, explaining how and why the behavior of this structure differs from the other two. Additionally, we will examine possible influencing factors, such as material properties, structural deformation patterns, and energy absorption characteristics, to ensure a comprehensive analysis that meets the reader’s expectations.

We appreciate your suggestion and will incorporate these improvements in the revised manuscript. Please let us know if there are specific aspects you would like us to elaborate on further.

Reviewer #2: 

Comment 1

The structural layout is not good enough and the readability is poor.

Response 1

Thank you for your constructive feedback. We acknowledge the need to improve the structural layout and enhance the readability of the manuscript. We will carefully revise the organization of sections to ensure a clear and logical flow of information, making it easier for readers to follow the study. Additionally, we will refine the text to improve clarity, conciseness, and coherence, ensuring that complex concepts are presented in a structured and accessible manner.

We appreciate your suggestion and will make the necessary improvements accordingly. Please let us know if you have specific recommendations regarding section organization or formatting preferences.

Comment 2

 Innovation needs to be emphasized. Is it just a matter of different application scenarios? Is there any innovation in the corrugated structure itself?

Response 2

Thank you for your valuable feedback. We recognize the importance of emphasizing innovation in the study. In the revised manuscript, we will explicitly highlight the novel contributions of our work, clarifying whether the innovation lies solely in its application to different scenarios or if there are fundamental advancements in the corrugated structure design itself.

If our research introduces new geometric configurations, improved energy absorption mechanisms, or enhanced structural efficiency, we will make these aspects more explicit. Additionally, we will compare our approach with existing studies to better illustrate its uniqueness and contribution to the field.

We appreciate your insightful comments and will ensure that the innovation aspect is clearly articulated and well-supported in the revised version. Please let us know if there are specific areas where you would like more details.

Comment 3

The ANSYS version should be provided. I didn't see the introduction and steps of calculation by using ANSYS in the article, please provide detailed information.

Response 3

Thank you for your valuable feedback. We acknowledge the importance of specifying the ANSYS version used in the study and will ensure that this information is clearly stated in the revised manuscript. Additionally, we will provide a detailed explanation of the calculation process in ANSYS, including the model setup, boundary conditions, meshing strategy, material properties, solver settings, and post-processing methods. This will enhance the clarity and reproducibility of our work.

We appreciate your suggestion and will incorporate these improvements accordingly. If there are specific aspects of the ANSYS simulation process that you would like us to elaborate on further, please let us know.

Comment 4

What is the purpose of the experiment listed in the article, and is it to verify the calculation results? I did not see the verification process and results. If it's not for verification, is it using experiments as a means of research? So what is the connection with the previous calculation, and what is the logic behind putting it into a paper?

Response 4

Thank you for your insightful feedback. We recognize the need to clearly articulate the purpose of the experiment and its relationship with the calculation results. In the revised manuscript, we will explicitly state whether the experiment serves as a verification of the numerical simulations or if it is conducted as an independent research method to explore specific aspects of the study.

If the experiment is meant for verification, we will provide a clear comparison between experimental and numerical results, including error analysis and discussions on potential discrepancies. If the experiment is instead used as a research tool, we will elaborate on its role in the study, its connection to the calculations, and the logical progression from numerical modeling to experimental validation or exploration.

We appreciate your thoughtful comments and will ensure that the logical framework of the study is well-structured and transparent in the revised version. Please let us know if there are specific aspects you would like us to further elaborate on.

All authors have read and approved the re-submission of the manuscript! If you have any questions, please let me know!

We are looking forward to hearing from you!

Sincerely yours,

Qian Liu

---

## [Decision Letter · Decision Letter 1]

6 Apr 2025

Optimization design of top beam energy absorbing member of corrugated hydraulic support with three kinds of sandwich structures based on adaptive response surface method

PONE-D-24-57458R1

Dear Dr. Liu,

We’re pleased to inform you that your manuscript has been judged scientifically suitable for publication and will be formally accepted for publication once it meets all outstanding technical requirements.

Kind regards,

Yuan-Fong Chou Chau

Academic Editor

PLOS ONE

Additional Editor Comments (optional):

The manuscript has been thoroughly revised and is now suitable for publication.

Reviewers' comments:

Reviewer's Responses to Questions

**Comments to the Author**

1. If the authors have adequately addressed your comments raised in a previous round of review and you feel that this manuscript is now acceptable for publication, you may indicate that here to bypass the “Comments to the Author” section, enter your conflict of interest statement in the “Confidential to Editor” section, and submit your "Accept" recommendation.

Reviewer #1: All comments have been addressed

2. Is the manuscript technically sound, and do the data support the conclusions?

Reviewer #1: Yes

3. Has the statistical analysis been performed appropriately and rigorously? 

Reviewer #1: Yes

4. Have the authors made all data underlying the findings in their manuscript fully available?

Reviewer #1: Yes

5. Is the manuscript presented in an intelligible fashion and written in standard English?

Reviewer #1: Yes

6. Review Comments to the Author

Reviewer #1: (No Response)

7. PLOS authors have the option to publish the peer review history of their article (what does this mean? ). If published, this will include your full peer review and any attached files.

**Do you want your identity to be public for this peer review?** For information about this choice, including consent withdrawal, please see our Privacy Policy .

Reviewer #1: No

---

## [Editor Report · Acceptance letter]

PONE-D-24-57458R1

PLOS ONE

Dear Dr. Liu,

I'm pleased to inform you that your manuscript has been deemed suitable for publication in PLOS ONE. Congratulations! Your manuscript is now being handed over to our production team.

Kind regards,

on behalf of

Dr. Yuan-Fong Chou Chau

Academic Editor

PLOS ONE